# The Effect of a Single Bout of Chinese Archery on Core Executive Functions in Preadolescent Children in Shanghai

**DOI:** 10.3390/ijerph20021415

**Published:** 2023-01-12

**Authors:** Jianjun Liu, Surasak Taneepanichskul, Bo Zhang, Peter Xenos

**Affiliations:** 1School of Physical Education & Health, Shanghai University of International Business and Economics, Shanghai 201620, China; 2College of Public Health Sciences, Chulalongkorn University, Bangkok 10330, Thailand; 3Capitol University Research and Extension Office (CUREXO), Capitol University, Cagayan de Oro City 9000, Philippines

**Keywords:** Chinese archery, mind-body unit, core executive function, preadolescent children

## Abstract

Background: Recent literature has demonstrated that acute physical activity benefits the transfer of executive functions. However, further studies indicated the type of variability in the effect of physical activity on executive functions. Therefore, this study initially explored the effect of a single bout of Chinese archery on subdomains of core EFs in preadolescent children; Method: Eligible participants were allocated either an intervention group (*n* = 36) or a control group (*n* = 36). The subjects in the intervention group received a 45-min Chinese archery session. The primary outcomes were the performance of core EFs (inhibition control, working memory, and cognitive flexibility) in preadolescent children, assessed with psychological paradigms (Fish Flanker Task, N-Back Task, and DCCS (Border version), respectively). Paired *t*-test and ANCOVA were used to analyze the mean difference in the performance of core EFs within and between subjects, respectively; Results: Considering reaction time and accuracy, we explored the impressive performance in three cognitive tasks with acute Chinese archery; Conclusion: The finding suggests that a single bout of Chinese archery benefited three subdomains of core EFs in healthy preadolescent children.

## 1. Introduction

An emerging body of evidence has verified the positive association of executive functions (EFs) induced by physical activity programs throughout one’s lifespan [1,2,3,4,5,6]. EFs involve those cognitive processes that underlie goal-directed behavior and are coordinated by activity within the prefrontal cortex [7,8,9]. EFs vitally influence individuals throughout their lifespans in areas such as mental and physical health [10], academic achievements [11], career success [12], and social and psychological development [13]. Generally, It is accepted that core EFs are composed of inhibition control, working memory, and cognitive flexibility [8].

Although most studies in this area have emphasized chronic physical activity program engagement, there is an emerging body of research on EFs following a single bout of physical activity program [3,14,15,16]. Conclusively, a single bout of physical activity program positively influences EFs [17,18,19]. The Stroop task has verified the most significant enhancement (ES: 0.2–1.16) of inhibition control following acute physical activity program [20]. However, there is less literature and evidence regarding the positive effect on working memory and cognitive flexibility.

Furthermore, there is limited knowledge on how to impact core EFs with acute physical activity program cessation across one’s lifespan [21]. Yet, most literature has highlighted young adults and elderly populations, with less research on other diverse subpopulations across their lifespans [22]. Importantly, this is a well-developed period for core EFs, which are sensitive to physical activity during childhood [8]. Meanwhile, EFs in early life have been predicted for achievement, health, wealth, and quality of life throughout one’s lifespan [23]. Therefore, it is highly relevant to call for extra investigation addressing EFs enhancement following a single bout of physical activity program in the school-aged population.

What acute physical activity program characteristics generate these effects? Previous literature has asserted the primary moderators to be the intensity, duration, and type of physical activity [14,24,25]. Most of the research has highlighted the association between the intensity of a single bout of physical activity program and its effect on EFs. Generally, moderate physical intensity is prominently supported as the optimal intensity to improve EFs with the inverted U-shape theory in dose-response relation [14]. Conversely, the evidence observed the most significant enhancement of EFs compared to moderate physical intensity [26,27,28,29,30,31]. Consequently, it is premature to conclude that any passion is superior.

Although the evidence has optimized 16–35 min as the duration of a single bout of physical activity at present [14,24], the critical threshold which induces the transfer of EFs is not indicated [25]. Moreover, there may be other necessary dependencies, such as personal tolerance, steady-state activity, and familiarity with physical stimuli [20,32]. Meanwhile, the dose of a single bout of physical activity program is proposed as a more precise characteristic variable linked to the enhancement of EFs following the relevant intervention.

As a critical question, increasing evidence demonstrates that there are not equal improvements in EFs associated with different types of physical activity [1,4,33,34,35,36]. Indeed, Lambourne and Tomporowski [37] confirm that a single bout of cycling (effect size = 0.23) achieved considerably more enhancement of cognition than running-based activities (effect size = 0.12). Another recent study shows that physical activity with cognitive demands reaped more enhancement [38,39,40]. Accordingly, the typical characteristics of physical activity that might obtain the most remarkable EFs enhancement are valuable to be clarified. Physical activities’ cognitive, social, emotional, coordinated, and environmental load warrants critical consideration [33,41,42,43]. However, there is a lack of more diverse types of physical activity implemented to enhance EFs apart from repetitive aerobic-type activity [14]. More diverse types of acute physical activity programs have been called for, especially those with mindfulness activities [44,45].

As one of the representatively traditional (sports) martial arts in China, Chinese archery originated from hunting activities with bow and arrow in the Stone Age and developed from archery rituals, which have been regarded as one of the six essential skills required of pupils since the West Zhou Dynasty (1046 BC-771 BC). With the evolution of history, Chinese archery has been described as a spiritual (expression of one’s inner self) and mental (emotion release) approach in the ritual setting [46].

Chinese archery emphasizes mind-body unity and pursues simultaneous development of technique and Tao of the individual [47]. Hence, it is regarded to strengthen two primary advantages. Firstly, it strengthens one’s physical health, such as muscle strength of the back and arm [48], control of respiration, heart [49], arousal system [50], and motor execution [51]. Secondly, it strengthens one’s mind cultivation, such as self-reflection, self-regulation, concentration, discipline, and individual character development (e.g., humility, virtue, respect, morality, Tao philosophy) [47].

The Chinese government is revitalizing and promoting traditional martial arts programs such as Chinese archery in primary and secondary schools to promote the development of children and adolescents. So far, the public health benefit of Chinese archery is mainly based on historical and expert experience. There are few relevant experimental supports.

Consequently, this study initially explored the effects of a single bout of Chinese archery on subdomains of core EFs in preadolescent children in Shanghai. We hypothesized that a single bout of Chinese archery significantly improves core EFs in preadolescent children in Shanghai.

## 2. Materials and Methods

### 2.1. Participants

All participants were recruited from those students in 4th grade in the Songjiang Experimental School Affiliated with Shanghai University of International Business and Economics (SUIBE) in Shanghai, China. The inclusion criteria in this study are right-hand dominant, regular physical fitness and health conditions, age ranges from 9 to 11 years, and no history of neurological and psychiatric disease.

Eventually, 72 inclusive participants were allocated either an active control group (*n* = 36) or an interventional group (*n* = 36) based on intention. The Declaration of Helsinki conducted this study with the approval of the ethics review committee for research at Chulalongkorn University (Protocol NO. 073. 1/64). All participants and their parents were informed and assigned consent before participation.

### 2.2. Experimental Design

The study incorporates three sessions with different periods (Figure 1). Firstly, during the screening session (*t*_0_), demographic data were collected. Using a between-subjects pretest posttest comparison design, all eligible subjects were assigned to receive either intervention group or active control group. Secondly, all subjects completed the core EFs performance assessment as pretest (*t*_1_) before treatment. Thirdly, after a 45-min treatment period, posttest (*t*_2_) was conducted in both groups. Core EFs performance assessment includes inhibition control, working memory, and cognitive flexibility. Subjects in the intervention group performed Chinese archery practice. The subjects in the active control group performed regular extracurricular activities.

### 2.3. Measurements

Demographic data were collected from the physical fitness assessment database in school in the current semester.

Core Executive Functions

Three computer-based neuropsychological paradigms (Fish flanker task [52], N-Back test [53], and DCCS (Border version) [54] were applied to evaluate the performance of inhibition control, working memory, and cognitive flexibility, respectively. E-prime software 2.0 (Psychology Software Tools Inc., Pittsburgh, PA, USA) carried out the relevant paradigms test and data collection. All assessments were performed in the quiet computer room. After the intervention activity, all subjects were required to return to the same computer room within 10 min and complete the same cognitive task assessments after a few minutes of rest. The reaction times of correct responses and the accuracy were collected as the evaluation index of core EFs performance.

#### 2.3.1. Inhibition Control

The Fish Flanker task was applied to evaluate the inhibition control of participants [3,55].

Material: Two different trial types were presented, congruent and incongruent. In each trial, subjects tried to offer fast and accurate responses when determining the direction of the center fish. In congruent trials, all five fish in the stimulus array were revealed to be facing the same direction. In incongruent trials, the four distracting fish were indicated in the opposite direction of the central target fish. Regarding size, each fish stimulus was displayed as 3 cm tall and separated by 1 cm or a visual angle of 1.72°. The target fish was always located in the exact location in every trial. The stimuli were presented for 1.5 s until a response was for each trial. A new trial was presented after an interval (“+” with blue background) of 0.5 s.

Procedure: Before each formal trial, each block of practice trials (12 congruent trials and 12 incongruent trials) was conducted to ensure sufficient accuracy for all participants. Formal trials consisted of 48 congruent trials and 48 incongruent trials, respectively, followed by 1-min intervals between 2 block trials. All trial types were randomly presented with equal probability.

#### 2.3.2. Working Memory

The N-back task (*n* = 1, 2) in this study was used to evaluate the working memory performance of participants.

Materials: Two different conditions (n back, *n* = 1, 2) were performed, and each condition was presented in a fixed order for all participants. Six different shapes with various colors (green circle, red line, blue cross, purple star, brown square, and orange triangle) were displayed focally in an assigned manner. The figure of shapes was a height of 3 cm. Each trial was presented for 2.9 s on a black background, then 3 s with a fixed inter-stimulus.

Procedure: All participants were administered to respond (L button press meant the same matching target. Otherwise, the A button press was determined) whether the present stimulus matched the previous shape displayed *n* (*n* = 1, 2 in the present study) steps before the target in a sequence of shape stimuli. During the task, all participants were demonstrated to respond via visual stimuli as fast and accurately as possible with an assigned button (L or A). Formal trials consisted of 2 condition block trials (24 1-back trials and 24 2-back trials), followed by a 1-min interval between 2 block trials. Before each formal trial, each block of training trials (10 1-back trials and 10 2-back trials) was conducted to ensure sufficient accuracy for all participants during the formal experiment.

#### 2.3.3. Cognitive Flexibility

DCCS (Border version) was implemented to assess participants’ cognitive flexibility in the present study. The target cards of various colors (blue and red) and shapes (boat and rabbit) were utilized in the DCCS (Border version).

Materials: Three different trial types were presented: in color trials, the participants were required to sort test cards according to color; in shape trials, the shape was the cue of sorting test cards; and in border trials, if the black border test cards were displayed, color trials were executed. Otherwise, the shape trials had to be conducted. Participants were required to pass the color trials, shape trials, and border trials in turn. The stimuli of the DCCS (Border version) were 5 cm tall and 9 cm wide target cards (rabbit and boat with bivalent color). Each trial started when a 3-s stimulus was presented on a black background, followed by a fixed intertrial interval of 2.5 s.

Procedure: All subjects should try to give fast and accurate responses when visual stimuli are presented on the screen. For smooth operation during the trials, each phase of practice trials (8 color trials, 8 shape trials, and 12 border trials) was given. In turn, participants completed three blocks of heterogeneous trials (16 color trials, 16 shape trials, and 32 border trials). Each block test followed a 1-min interval. During the trials, stimuli switches were presented randomly, ensuring that the same target card could not be displayed on the screen sequentially.

### 2.4. Experimental Procedure

All participants attended extracurricular activities at Songjiang Experimental School Affiliated to SUIBE, with a 45-min session each. The two extracurricular activities conducted concurrently after the scheduled classes.

In the intervention group, the children were instructed to practice Chinese archery by coach, who is extensively experienced with teaching diverse subpopulations and contexts. All children had received five sessions of Chinese archery and met the basic requirements, including the primary technique, traditional etiquette, mind regulation skills, and historic and cultural features. All participants had a week interval before treatment condition. The Chinese archery was in a dedicated outdoor venue. The contents of the intervention program included techniques (standing posture, setting arrow, pushing and pulling the string, lifting the bow, drawing a full bow, collimation, shot, closing form, etc.), mind training (self-reflection, self-regulation, concentration, deep-breathing techniques, etc), and basic etiquette (pre-etiquette, post-etiquette, etiquette to target).

An intervention session comprised of a warm-up section (5 min of stretching and 3 min of mind conditioning), a Chinese archery section (about 34 min), and an end section (2 min of meditation and 1-min-deep breathing).

While treatment starts, participants line up and must be taken up bow and arrow with a peaceful mind and posture erect, then fully concentrate on shooting process. Simultaneously, the central theme of the indoctrination from Chinese archery is that it is not principally concerned with scoring hits but with a circle of practice. The exercise session followed the principle of Chinese archery. The whole intervention process is completed under the control and guidance of the coach.

As an active control group, the children in control group participated in regular extracurricular activities scheduled by school in the real world. In this session, children were required to walk on the track for 10 min and spent the rest of the time doing homework together in the classroom.

### 2.5. Statistical Analysis

All statistical analyses were conducted using SPSS 25 (IBM Corp., Armonk, NY, USA). Demographic, anthropometric, and physical fitness variables between groups were analyzed using independent samples *t*-test and Chi-squared test. Paired *t*-tests were conducted to investigate the statistical difference in reaction time and accuracy (N-back task, fish Flanker task, DCCS task) in each group on core executive function. Considering the purpose of increasing statistical power and reducing potential bias owing to the baseline imbalance, ANCOVA was used to analyze the significant difference in reaction time and accuracy from relevant assessment (N-back task, fish Flanker task, DCCS task) between groups, pre-test performance as covariates and post-test performance as the dependent variable. The estimated effect size was presented as partial eta square (η^2^). *p* < 0.05 was set as statistical significance.

## 3. Results

### 3.1. Demographic Data

The independent samples *t*-test and Chi-squared test revealed the detailed descriptive statistics of demographic, anthropometric, and aerobic fitness data for the subjects in Table 1. The outcomes showed that the height of subjects reached a considerable difference between groups (t = −2.19, *p* < 0.05).

### 3.2. Core Executive Functions Performance

The detailed descriptive statistics of EFs are displayed in Table 2.

#### 3.2.1. Inhibition Control

Regarding the reaction time, the Paired-samples T-test revealed that there was a significance (t =7.36, *p* = 0.00) in congruent trials in the intervention group, pre-test (558.95 ± 99.40) milliseconds compared to post-test (472.05 ± 47.18) milliseconds (Figure 2a). Similarly, a statistical difference (t = 3.50, *p* = 0.00) was also detected in the control group with pre-test (542.61 ± 91.90) milliseconds compared to post-test (504.31 ± 63.28) milliseconds (Figure 2a). In contrast, there was no significance in incongruent trials in both groups.

ANCOVA revealed statistical difference in congruent trials (F _(1,36)_ = 17.48, *p* = 0.00, η^2^ = 0.20) (Figure 2a) and incongruent trials (F _(1,36)_ = 9.15, *p* = 0.00, η^2^ = 0.12) (Figure 2b) between groups, respectively. Moreover, faster reaction time is demonstrated in the intervention group with two trials above.

Regarding accuracy, the Paired-samples T-test postulated that there is not significant difference presented in congruent trials (t = −0.60, *p* = 0.55) and in incongruent trials (t = −0.06, *p* = 0.95) in the control group. However, a significant difference (−7.05 ± 18.97), (95% CI: −13.47–−0.63) was detected in incongruent trials (t = −2.23, *p* = 0.03) with pre-test (86.56 ± 14.54) % compared to post-test (93.61 ± 9.62)% (Figure 3b) rather than in congruent trials (t = −1.09, *p* = 0.28) in the intervention group.

Meanwhile, significantly higher accuracy was revealed in congruent trials (F _(1,36)_ = 4.53, *p* = 0.04, η^2^ = 0.06) (Figure 3a) and incongruent trials (F _(1,36)_ = 5.85, *p* = 0.02, η^2^ = 0.08) between groups, using ANCOVA test (Figure 3b).

#### 3.2.2. Working Memory

Regarding the reaction time, the Paired-samples T-test displayed a significant difference (t = 3.12, *p* = 0.00) in 1-back trials with pre-test (1147.54 ± 287.80) milliseconds than post-test (1017.16 ± 249.75) milliseconds in the intervention group (Figure 4a). Additionally, with pre-test (1120.25 ± 300.12) milliseconds compared to post-test reaction time (843.33 ± 240.06) milliseconds, the significance was revealed in 2-back trials (t = 7.60, *p* = 0.00) (Figure 4b).

Furthermore, a significant difference (t = 2.18, *p* = 0.04) showed in 1-back trials in the control group, pre-test (1010.47 ± 308.65) milliseconds compared to post-test (899.33 ± 319.07) milliseconds (Figure 4a). In contrast, the difference failed to reach significance in 2-back trials (t = 1.92, *p* = 0.06).

In addition, ANCOVA revealed a significant difference in 2-back trials (F _(1,36)_ = 4.44, *p* = 0.04, η^2^ = 0.06) between groups (Figure 4b). The faster performance was shown in an intervention group. However, no statistical differences were detected in 1-back trials between groups (F _(1,36)_ = 0.63, *p* = 0.34, η^2^ = 0.00).

Regarding accuracy, the Paired-samples T-test demonstrated a significant improvement (−5.06 ± 13.81), (95% CI: −9.73–−0.38) in the 2-back trials (t = −2.20 *p* = 0.04), with pre-test (70.7 ± 9.2)% compared to post-test (75.7 ± 9.3)% in the intervention group (Figure 5b), although it did not reach significance in 1-back trials (t = −1.93, *p* = 0.06). Moreover, a significant decline was displayed in 2-back trials (t = 2.23, *p* = 0.03) with pre-test (67.3 ± 11.4)% compared to post-test (62.4 ± 14.0)% in the control group (Figure 5b).

Additionally, ANCOVA displayed a significant difference in 1-back trials (F _(1,36)_ = 9.1, *p* = 0.00, η^2^ = 0.12) (Figure 5a) and 2-back trials (F _(1,36)_ = 19.9, *p* = 0.00, η^2^ = 0.22) (Figure 5b) between groups. The intervention group had higher accuracy.

#### 3.2.3. Cognitive Flexibility

Regarding reaction time, the Paired-samples T-test found significant differences in color trials (t= 3.03, *p* = 0.01) with pre-test (768.70 ± 165.50) milliseconds compared to post-test (707.87 ± 123.19) milliseconds (Figure 6a), shape trials (t = 2.19, *p* = 0.04) with pre-test (694 ± 176.09) milliseconds compared to post-test (640.59 ± 138.60) milliseconds (Figure 6b), and the border trials (t = 6.64, *p* = 0.00) with pre-test (1487.79 ± 319.05) milliseconds compared to post-test (1133.81 ± 241.82) milliseconds (Figure 7a) in the intervention group, separately. In contrast, no significant difference was detected in the color trials (t = 1.74, *p* = 0.09), shape trials (t = -.067, *p* = 0.51), and border trials (t = 1.56, *p* = 0.13) in the control group.

Likewise, ANCOVA revealed a statistical shorter reaction time in the color trials (F _(1,36)_ = 6.18, *p* = 0.02, η^2^ = 0.08) (Figure 6a), shape trials (F _(1,36)_ = 7.64, *p* = 0.00, η^2^ = 0.10) (Figure 6b), and border trials (F _(1,36)_ = 6.16, *p* = 0.02, η^2^ = 0.08) (Figure 7a) between groups, respectively.

Regarding accuracy, the Paired-samples T-test found considerable differences (−7.24 ± 8.08), (95% CI: −9.97–−4.51) in the border trials (t = −5.37, *p* = 0.00) with pre-test (63.2 ± 8.9)% compared to post-test (70.4 ± 9.4)% (Figure 7b). However, a significant difference was not detected in color trials (t = −0.68, *p* = 0.50) and shape trials (t = 0.11, *p* = 0.92). Similarly, no significant effects were identified in the color trials (t = −1.53, *p* = 0.14), shape trials (t = 1.00, *p* = 0.32), and border trials (t = −0.93, *p* = 0.36) in the control group.

Meanwhile, ANCOVA found significant differences in border trials (F _(1,36)_ = 13.4, *p*= 0.00, η^2^ = 0.16) between groups; the intervention group showed higher accuracy (Figure 7b). Controversy, no considerable difference was presented in color trials (F _(1,36)_ = 0.13, *p* = 0.72, η^2^ = 0.00) and shape trials (F _(1,36)_ =1.1, *p* = 0.30, η^2^ = 0.02).

## 4. Discussion

This study intends to explore the influence in subdomains performance of core EFs after a single bout of Chinese archery in preadolescent children. Subsequently, our results support the hypotheses that a single bout of Chinese archery significantly improves preadolescent children’s inhibition control, working memory, and cognitive flexibility. Specifically, the considerably faster response reaction time was confirmed in congruent trials, 2-back trials, and border trials within and between subjects. In addition, regarding accuracy, the impressive higher in congruent and incongruent trials, 2-back trials, and border trials were consistently detected within and between subjects.

Over the last decade, investigations into the transient effect of a single bout of physical activity have been mainly focused on the inhibitory control subdomain of core EFs, with a prominent 41% of the published studies [14]. Commonly, the outcomes have supported that an single bout of moderate-intensity [56] or HIIT [28,57] physical activity positively impacts inhibitory control in children regardless of the assessment task employed. For example, Hillman et al. [3] declared that a single 20-min bout of moderate-intensity aerobic exercise improves performance in incongruent trials in healthy preadolescents. Likewise, Cooper et al. [58] observed a significant effect on response times of the simple Stroop test (R: 818 ± 33 ms, E: 772 ± 26 ms; *p* = 0.03) and the complex level of the Stroop test (R: 1095 ± 36 ms, E: 1043 ± 37 ms; *p* = 0.04) in 44 adolescents (12 ± 0.6 year) following 10 min high-intensity sprint-based exercise. Consequently, a small-to-moderate effect size is found regarding the effect of a single bout of moderate-intensity physical activity on inhibitory control in healthy preadolescents (6–12 years) [Hedges’ g = 0.28; CI: 0.01, 0.56; *p* = 0.04] [59]; similar conclusions on HiiT physical activity are supported [60,61]. Our outcome in the current study is consistent with previous literature.

Compared with inhibition control, there is increasing attention being paid to demonstrating the effect of a single bout of physical activity on working memory performance [16,26,62,63] and cognitive flexibility performance [26,41,64,65] in the preadolescent subpopulation. Specifically, Ishihara et al. [66] announced that working memory and inhibitory control achieved the best benefit after 50-min game-based tennis lessons compared to technique-based lessons and watching TV. Using a between-subjects pretest-post-test design, working memory and inhibitory control were assessed after 15 min of intervention. Additionally, Chen et al. [15] initially reported the impressive benefits of a 30-min group jog at moderate intensity on inhibition, working memory, and the shifting of EFs in healthy preadolescent children (*n* = 83, F = 42, M = 41). Using a within-between subjects’ pretest-post-test design, three subdomains of core EFs were evaluated after 20–25 min of treatment.

Recently, relevant meta-analytic literature demonstrate a distinct benefit on working memory (SMD = −0.72; CI: −0.89, −0.56; *p* < 0.001) [17] and cognitive flexibility (SMD = −0.34; CI: −0.55, −0.14; *p* < 0.005) under acute moderate-intensity physical activity [67]. In contrast, a divergent conclusion has been postulated in previous meta-analyses [5,18]. The inconsistent findings are largely explained by the disparity of studies examining the effects of a single bout of physical activity on inhibitory control greatly exceed those assessing working memory and cognitive flexibility [14]. Indeed, as the literature advances, the type of physical activity program is probably regarded as a primary contributor that induces the transfer of EFs [1,68]. Contreras-Osorio et al. [69] illustrate a large effect size in all subcomponents of EFs on sports programs: working memory (ES = −1.25; CI: −1.70, −0.79; *p* < 0.00); inhibitory control (ES = −1.30; CI: −1.98, −0.63; *p* < 0.00); and cognitive flexibility (ES = −1.52; CI: −2.20, −0.83; *p* < 0.00). Moreover, Diamond and Ling review that mindfulness movement activity (such as yoga [45], Tai Chi [70,71,72], Chinese mind-body practices [73], and Quadrato motor training [74]) shows the most robust results to benefit EFs among all the ways (computer and noncomputer cognitive training, neurofeedback, school programs, physical activities, mindfulness practices, and miscellaneous approach) [75] in 179 studies. Interestingly, the outcome of examining three subdomains of core EFs in the present study positively corroborate previous literature.

Given the characteristics of Chinese archery, it appears to possess at least three advantages that enhance EFs in the present study. Firstly, it aims to reduce stress and foster mind-body unity to encourage positive moods and relaxed brain states. EFs rely on the prefrontal cortex and other neural regions, which become the most vulnerable with excess dopamine when suffering from negative emotion (e.g., stress, sadness, loneliness) [76,77]. Conversely, the reduction of stress [78,79] and positive mood [80] contribute to the enhancement of EFs. Bigelow et al. [81] illustrate that a 10-min session of mindfulness meditation displays a premium improvement (d = 0.55–0.86) of all executive function tasks (Stroop Task, TMT-B, and Leiter-3) compared to 10 min of exercise with a within-subjects pretest-post-test design in children (*n* = 16, Mean_age_ = 11.38 (±1.5)) with ADHD. During the practice of Chinese archery, participants must fully concentrate on motor coordination and breathing rhythm in a calm and relaxed mood. The state facilitates the arousal of the mind-body unity state as soon as possible. It produces a simultaneous concentrated, relaxed state of the brain, which promotes executive function performance [34].

Secondly, over-competition in sports can destroy self-esteem and character development. Participants during Chinese archery practice must take up the bow and arrow with a peaceful mind, erect posture, and a fully concentrated shooting process [82]. Chinese archery is not principally concerned with scoring hits because participants are not matched in their strengths. Those participants are required to highlight comparing one’s past and peer interaction instead of scoring hits [47]. This process produces less aggression and anxiety. Conversely, more self-esteem is achieved. The effect facilitates the cultivation of introspection, self-regulation, and character development of individuals than competitive sports [47].

Thirdly, the key to physical activity-induced benefits of EFs is that the diverse and continuous challenges of EFs are addressed in the real world [66]. Indeed, the exercise-induced transfer of EFs might be determined by the complex, controlled, and adaptive cognition and movement of the degree addressed in exercise [68]. During Chinese archery performance, participants would be confronted with the diverse situations in the real world. Those conditions push participants to continuously challenge various executive skills [82] that are conducive to improving executive function.

For instance, participants must bear complex motor sequences and discipline in mind, inhibit attending to distractions, and concentrate on every motor action during the process. Meanwhile, cognitive flexibility is also challenged; participants would constantly adjust their targeting actions and timing of shooting arrows based on their experience and prediction of unexpected changes in a natural setting.

The exercise protocol in the present study utilized the real world with ecological validity, which is based on extracurricular activities (ECA) in school, rather than most of the investigations being conducted in the laboratory with treadmills or cycle ergometers [14]. Additionally, ECA in school provides children with more opportunities for social interaction and a better motivational climate. With growing concerns about the younger generation’s health, ECA-based programs in school have several critical advantages (e.g., economy, convenience, breadth, diversity, and accessibility). Importantly, this study not only objectively proves that Chinese archery significantly improves core EFs in adolescent children but also further extend current knowledge how physical activities are the most effective way to enhance executive function [83]. Furthermore, the encouraging finding provides a highly potential application in public health promotion in normal preadolescent children and education practice on a large scale.

There are several limitations to this study. First, the randomized trial was not conducted in the present study but was a quasi-experimental design. Therefore, it is possible to produce more bias in the results of this study. Therefore, it is necessary to consider future research involving a randomized controlled study design to reduce bias. Second, an objective measure of physical activity during intervention is missing. Regarding the role of intensity as moderator [24], it is acceptable to monitor the heart rate range in real-time during the intervention and collect data as a subjective index for all participants in the future. Third, although the real-world interventional setting is regarded as a strength in this study, it also inevitably increased other confounders. This study detected an impressive result in the benefit of inhibitory control, working memory and cognitive flexibility performance. However, the long-term effects of this physical activity program and the sustainability of its benefit remain unknown. Further investigations are valuable. Finally, the adolescent subpopulations with EF deficiencies or disorders merit further investigations in the future.

## 5. Conclusions

To the best of our knowledge, the current study is the first to experimentally examine the effect of Chinese archery on core EFs in preadolescent children. This study supports that a single bout of Chinese archery positively impacted core EFs in preadolescent children. Furthermore, compared with the control group, shorter reaction time and higher accuracy of subcomponents of core executive functions in the Chinese archery group were significantly detected. it indicates an encouraging application for improving core executive functions with Chinese archery practice in normal preadolescent children in large-scale educational setting.

## Figures and Tables

**Figure 1 ijerph-20-01415-f001:**
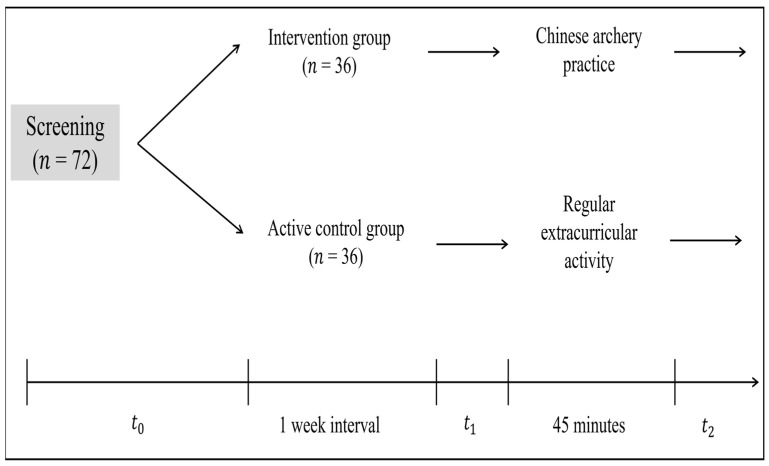
Experimental design. The study included three sessions (*t*_0_, *t*_1_, and *t*_2_) and two groups: an intervention group (*n* = 36) and an active control group (*n* = 36). The experimental group performed Chinese archery practice while the active control group was a regular extracurricular activity. The main outcome variables were core executive functions performance.

**Figure 2 ijerph-20-01415-f002:**
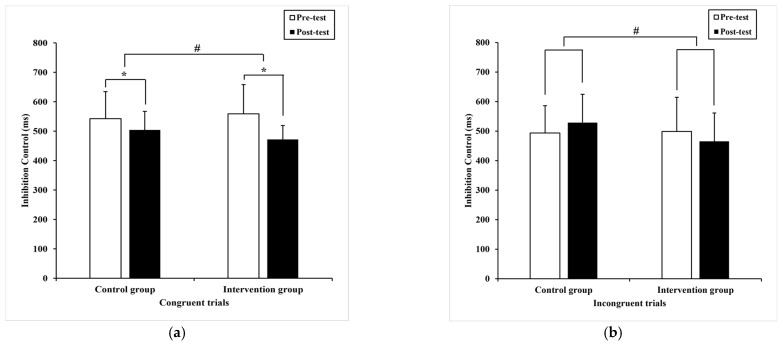
The performance of inhibition control within and between subjects. (**a**) the mean difference in reaction time of congruent trials; (**b**) the mean difference in reaction time of incongruent trials. Note, * present *p* < 0.05, within subjects and # present *p* < 0.05, between subjects, respectively. The data are presented as the mean ± SD.

**Figure 3 ijerph-20-01415-f003:**
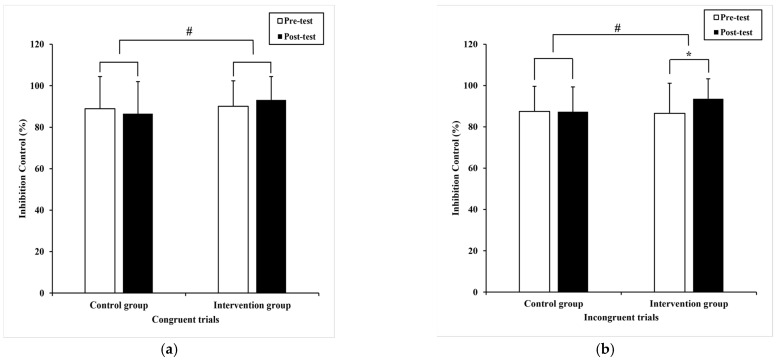
The performance of inhibition control within and between subjects. (**a**) the mean difference in accuracy of congruent trials; (**b**) the mean difference in accuracy of incongruent trials. Note, * present *p* < 0.05, within subjects and # present *p* < 0.05, between subjects, respectively. The data are presented as the mean ± SD.

**Figure 4 ijerph-20-01415-f004:**
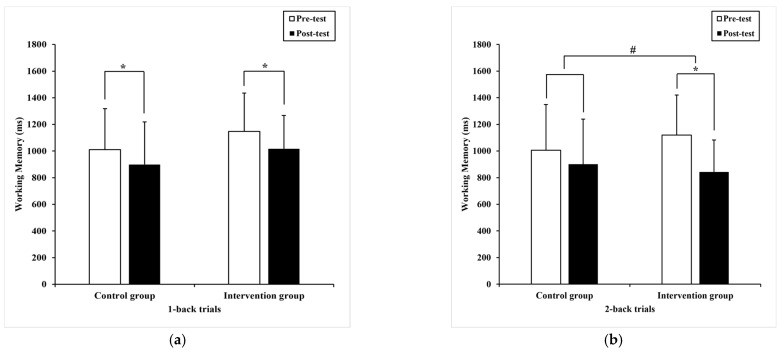
The performance of working memory within and between subjects. (**a**) the mean difference in reaction time of 1-back trials; (**b**) the mean difference in reaction time of 2-back trials. Note, * present *p* < 0.05, within subjects and # present *p* < 0.05, between subjects, respectively. The data are presented as the mean ± SD.

**Figure 5 ijerph-20-01415-f005:**
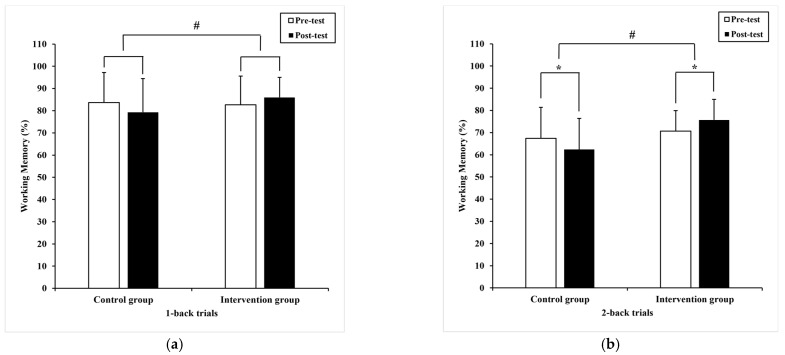
The performance of working memory within and between subjects. (**a**) the mean difference in accuracy of 1-back trials; (**b**) the mean difference in accuracy of 2-back trials. Note, * present *p* < 0.05, within subjects and # present *p* < 0.05, between subjects, respectively. The data are presented as the mean ± SD.

**Figure 6 ijerph-20-01415-f006:**
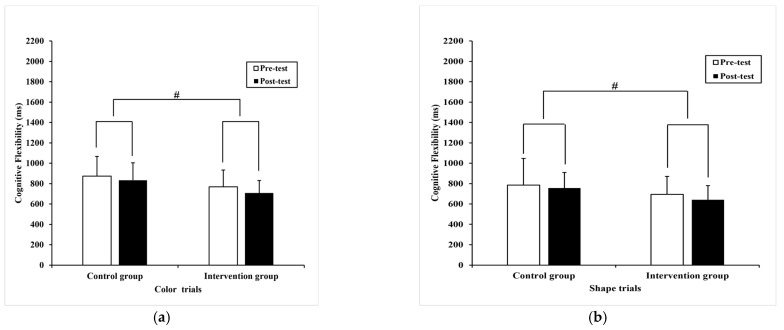
The performance of cognitive flexibility within and between subjects. (**a**) the mean difference in reaction time of color trials; (**b**) the mean difference in reaction time of shape trials. Note, # present *p* < 0.05, between subjects, respectively. The data are presented as the mean ± SD.

**Figure 7 ijerph-20-01415-f007:**
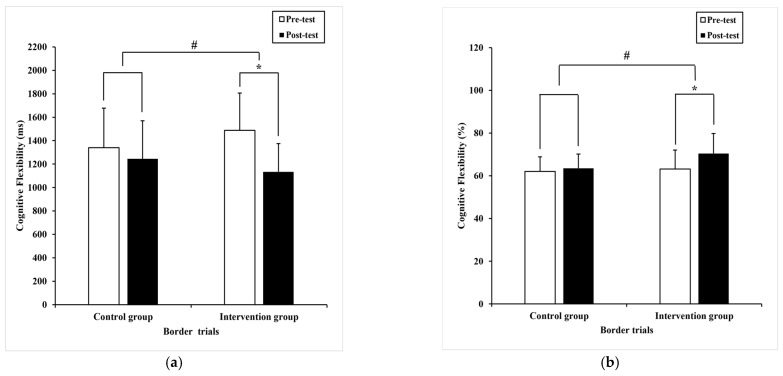
The performance of cognitive flexibility within and between subjects. (**a**) the mean difference in reaction time of border trials; (**b**) the mean difference in accuracy of border trials. Note, * present *p* < 0.05, within subjects and # present *p* < 0.05, between subjects, respectively. The data are presented as the mean ± SD.

**Table 1 ijerph-20-01415-t001:** Participants’ demographics and aerobic fitness.

Variables	Control Group	Intervention Group	t/x^2^	*p*
	M (SD)	M (SD)		
*n* (male/female)	36 (15/21)	36 (23/13)	3.57	0.06
Age (month)	115.94 (3.46)	115.44 (3.23)	0.63	0.53
Height (cm)	143.76 (6.28)	147.07 (6.56)	−2.19	0.00 *
Weight (kg)	41.90 (11.56)	42.01 (9.59)	−0.04	0.97
BMI (kg/m^2^)	20.07 (4.47)	19.21 (3.07)	0.95	0.35
Aerobic fitness (ml/min/kg)	46.08 (2.74)	45.20 (2.65)	1.39	0.17

Note. *n* present number of participants; BMI= Body mass index; Aerobic fitness is presented by assessment of VO_2max_; * mean *p* < 0.05 between subjects.

**Table 2 ijerph-20-01415-t002:** Performance from subcomponents of core executive function, data presented as mean (SD).

	Control Group (*n* = 36)		Intervention Group (*n* = 36)		F	*p*	η^2^_p_
	Pre-Test M (SD)	Post-Test M (SD)	Pre-Test M (SD)	Post-Test M (SD)			
Reaction time (ms)							
Inhibition Control							
Congruent trials	542.61 (91.90)	504.31 * (63.28) 507.71 ^a^ (6.59)	558.95 (99.40)	472.05 * (47.18) 468.66 ^a^ (6.59)	17.48	0.00 ^#^	0.20
Incongruent trials	493.96 (92.28)	529.08 (95.40) 529.81 ^a^ (15.24)	498.98 (115.94)	465.33 (95.99) 464.60 ^a^ (15.24)	9.14	0.00 ^#^	0.12
Working Memory							
1-back trials	1010.47 * (308.65)	899.33 (319.07) 935.00 ^a^ (40.96)	1147.54 (287.8)	1017.16 * (249.75) 981.48 ^a^ (40.96)	0.62	0.34	0.01
2-back trials	1005.43 (342.97)	901.02 (338.57) 932.13 ^a^ (39.91)	1120.25 (300.12)	843.33 * (240.06) 812.22 ^a^ (39.91)	4.44	0.04 ^#^	0.01
Cognitive Flexibility							
color trials	874.12 (194.15)	833.60 (170.65) 802.56 ^a^ (18.12)	768.77 (165.50)	707.87 * (123.19) 738.05 ^a^ (17.86)	6.18	0.02 ^#^	0.08
shape trials	784.37 (263.64)	755.96 (153.89) 742.30 ^a^ (22.46)	694.36 (176.09)	640.59 * (138.60) 654.22 ^a^ (22.14)	7.64	0.01 ^#^	0.10
Border trials	1339.90 (336.90)	1245.0 (324.87) 1269.39 ^a^ (45.00)	1487.79 (319.05)	1133.81 * (241.83) 1109.43 ^a^ (45.00)	6.16	0.02 ^#^	0.08
Accuracy (%)							
Inhibition Control							
Congruent trials	88.9 (15.5)	86.52 (15.47) 86.46 ^a^ (2.26)	90.06 (12.32)	93.19 (11.23) 93.26 ^a^ (2.26)	4.53	0.04 ^#^	0.06
Incongruent trials	87.5 (12.1)	87.29 (12.07) 87.34 ^a^ (1.82)	86.56 (14.54)	93.61 * (9.62) 93.57 ^a^ (1.82)	5.85	0.02 ^#^	0.08
Working Memory							
1-back trials	83.7 (13.5)	79.30 (15.18) 79.02 ^a^ (1.70)	82.68 (12.87)	85.98(8.97) 86.27 ^a^ (1.70)	9.07	0.00 ^#^	0.12
2-back trials	67.4 (14.0)	62.41 * (14.03) 62.91 ^a^ (1.94)	70.65 (9.24)	75.71 * (9.28) 75.21 ^a^ (1.94)	19.89	0.00 ^#^	0.22
Cognitive flexibility							
Color trials	90.2 (20.0)	95.5 (6.62) 95.49 ^a^ (1.29)	91.35 (18.71)	93.61 (8.69) 93.59 ^a^ (1.29)	1.07	0.30	0.02
Shape trials	93.4 (8.5)	90.4 (18.4) 90.4 ^a^ (3.0)	92.29 (16.78)	91.95 (17.85) 91.9 ^a^ (3.0)	0.131	0.72	0.00
Border trials	62.1 (6.8)	63.53 (6.66) 63.77 ^a^ (1.24) 59.6 ^a^ (1.0)	63.19 (8.91)	70.43 * (9.41) 70.19 ^a^ (1.24) 65.5 ^a^ (1.0)65.6 * (6.5) 65.5 ^a^ (1.0)	13.40	0.00 ^#^	0.16

Note: * mean *p* < 0.05 within subjects, ^#^ mean *p* < 0.05 between subjects, ^a^ mean adjusted value.

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
