# Peer review of "The Effect of a Single Bout of Chinese Archery on Core Executive Functions in Preadolescent Children in Shanghai"

_ijerph, 2023, doi:10.3390/ijerph20021415_

Round 1

Reviewer 1 Report

Introduction: A little bit long and wordy. I suggest you to revise it in a more concise way

Methods: The intervention group performed 34 minutes of Chinese Archery while the control group only 10 of walking. Please explain this.

Author Response

Dear Professor,

On behalf of my co-authors, we appreciate you very much for your positive and constructive comments and suggestions on our manuscript. We have studied the reviewers’ comments carefully. We have tried our best to revise our manuscript and provide point-by-point responses according to your comments. The following is a detailed list of responses:

Point 1: Introduction: A little bit long and wordy. I suggest you to revise it in a more concise way

Response 1: According to your recommendation, we deleted the sentence from Lines 74 to 76, and the sentence from Lines 85 to Line 88, and then, the introduction is more concise and smooth than before the version.

Point 2: Methods: The intervention group performed 34 minutes of Chinese Archery while the control group only 10 of walking. Please explain this.

Response 2: The control group in this study is an active control group. All activities in the control group were carried out according to the regular extracurricular activities scheduled by the school. In China, extracurricular activity always contains light to moderate-intensity physical activity programs and sedentary behaviour. In this session, the children were given a 10-minute walk on the track and then complete their homework together in their classroom. We just compared Chinese archery practice with regular extracurricular activity. It is not a completed active control group but a condition in the real world.

Reviewer 2 Report

The aim of this study was to investigate the effect of a single bout of Chinese archery on subdomains of core EFs in preadolescent children in Shanghai. The study is well-conducted, easy to read, and provides new insights into the exercise-cognition interaction, with particular interest on the acute effect of exercise on cognitive performance. I would like to congratulate the Authors for their work. Apart from a substantial language revision, I have only some minor comments that I hope will be useful to improve the overall quality of the manuscript.

Line 35-39. Physical exercise and physical activity are similar but not identical concepts, as each has a specific meaning. Please pay attention to this.

Line 49. Rephrase the question. Or consider to avoid it.

Line 76-77. Please cite previous studies on acute effect of other forms of exercise on cognitive performance, such as balance and yoga:

Formenti, D., Cavaggioni, L., Duca, M., Trecroci, A., Rapelli, M., Alberti, G., Komar, J., Iodice, P., 2020. Acute Effect of Exercise on Cognitive Performance in Middle-Aged Adults: Aerobic Versus Balance. J. Phys. Act. Health 17, 773–780. https://doi.org/10.1123/jpah.2020-0005

Gothe, N., Pontifex, M.B., Hillman, C., McAuley, E., 2013. The Acute Effects of Yoga on Executive Function. J. Phys. Act. Health 10, 488–495. https://doi.org/10.1123/jpah.10.4.488

Line 210. Control group performed an activity that is not comparable in terms of duration of physical activity to that of experimental group. This should be addressed and elaborated within the manuscript, as possible confounder of the results.

Methods. I suggest to add a Figure summarizing the experimental design of the study, thus allowing the Reader and immediate comprehension of the study.

Please pay attention to various typo within the manuscript.

In the conclusion section, I suggest to include possible practical applications derived from the study.

Author Response

Responses to Reviewer’s Comments

Dear Professor,

On behalf of my co-authors, we appreciate you very much for your positive and constructive comments and suggestions on our manuscript. We have studied the reviewers’ comments carefully. We have tried our best to revise our manuscript and provide point-by-point responses according to the comments. The following is a detailed list of responses:

Point 1: Line 35-39. Physical exercise and physical activity are similar but not identical concepts, as each has a specific meaning. Please pay attention to this. Line 49. Rephrase the question. Or consider to avoid it.

Response 1: We changed “physical exercise” and “physical activity” into “physical activity program” Lines No. 29, 35- 40, and Line No. 43, 49, 51, and 78.

Point 2: Line 76-77. Please cite previous studies on acute effect of other forms of exercise on cognitive performance, such as balance and yoga:

Formenti, D., Cavaggioni, L., Duca, M., Trecroci, A., Rapelli, M., Alberti, G., Komar, J., Iodice, P., 2020. Acute Effect of Exercise on Cognitive Performance in Middle-Aged Adults: Aerobic Versus Balance. J. Phys. Act. Health 17, 773–780. https://doi.org/10.1123/jpah.2020-0005

Gothe, N., Pontifex, M.B., Hillman, C., McAuley, E., 2013. The Acute Effects of Yoga on Executive Function. J. Phys. Act. Health 10, 488–495. https://doi.org/10.1123/jpah.10.4.48

Response 2: We cited the two new references (No. 49, 50) on Lines No. 76-77 in the manuscript. We deleted the references (No. 43, 79) and cited a reference (No. 50) in Line No. 402.

Point 3: Line 210. Control group performed an activity that is not comparable in terms of duration of physical activity to that of experimental group. This should be addressed and elaborated within the manuscript, as possible confounder of the results.

Response 3: The control group in this study is an active control group. All activities in the control group were carried out according to the regular extracurricular activities scheduled by the school. In China, extracurricular activity always contains low-intensity physical activity and sedentary behaviour. In this session, the children were given a 10-minute walk on the track and then complete their homework together in their classroom. We compared Chinese archery practice with regular extracurricular activity. It is not a completed active control group but a condition in the real world.

Point 4: Methods. I suggest to add a Figure summarizing the experimental design of the study, thus allowing the Reader and immediate comprehension of the study.

Response 4:

Response 4:

Figure 1. Experimental design. The study included three sessions (t0, t1, and t2) and two groups: an intervention group (n=36) and an active control group (n=36). The experimental group performed Chinese archery practice while the active control group was a regular extracurricular activity. The main outcome variables were core executive functions performance.

Point 5: Please pay attention to various typo within the manuscript.

Response 5: We carefully check typos in the manuscript and corrected the “ACONVA” into “ANCOVA” in Line NO. 267 and Line NO. 285, respectively, figure-2(d) into figure-7(b) in Line No.348, “t’ai chi” into “Tai Chi” in Line No. 402, and the “pushe” into “push” in Line NO. 436, deleted the “an” in Line No. 63.

Point 6: In the conclusion section, I suggest to include possible practical applications derived from the study.

Response 6: According to the result of this study, we concluded that it indicates an encouraging application for improving core executive functions with Chinese archery practice in normal preadolescent children in a large-scale educational setting.
